# Three Subtropical Species Adapt to Drought by Reallocating Biomass and Adjusting Root Architecture

**Zhenya Yang** [1,2,3], **Jiancheng Zhao** [1,3], **Huijing Ni** [1,3], **Hui Wang** [4] and **Benzhi Zhou** [2,*]

1   Zhejiang Provincial Key Laboratory of Bamboo Research, Zhejiang Academy of Forestry, Hangzhou 310023, China
2   Research Institute of Subtropical Forestry, Chinese Academy of Forestry, Hangzhou 311400, China
3   Northwest Zhejiang Bamboo Forest Ecosystem Positioning Observation and Research Station, National Forestry and Grassland Administration, Hangzhou 310023, China
4   Thousand-Island Lake Forest Farm of Chun'an County, Hangzhou 311700, China
*   Correspondence: bzzhou@caf.ac.cn

**Abstract:** The drought tolerance of plants is significantly influenced by their root architecture traits and root adaptive strategies, but the key root architecture traits that affect drought tolerance and the differences in drought adaptative strategies of species with varying root architectures are not yet clear. This study aimed to investigate the response of three species' roots to drought and evaluate the key root architecture traits affecting the drought tolerance of the three species. One-year-old potted seedlings of three species [Chinese fir (*Cunninghamia lanceolata* (Lamb.) Hook.), masson pine (*Pinus massoniana* (Lamb.)), and moso bamboo (*Phyllostachys edulis* (Carr.) H. de Lehaie f. edulis)] were planted in a greenhouse under three drought conditions (sufficient water supply, moderate drought, and severe drought) for 90 days. Biomass, root morphology [root surface area (RSA), root length (RL), root diameter (RD)], root architecture [root topological index (TI), fractal dimension (FD), and root branching angle (RBA)] of seedlings were measured monthly. The drought tolerance of species was quantified by studying the response ratio (RR) of root length and biomass in response to drought. We found that: (i) different levels of drought inhibited the biomass accumulation and root growth of the three species, and drought tolerance showed a decreasing order as pine > Chinese fir > bamboo; (ii) drought decreased the RD in bamboo but increased it in pine. Both bamboo and Chinese fir reduced their FD and RBA under drought stress, while pine was relatively stable. All the three species' roots tended to develop a herringbone branching architecture (increase their TI) under drought stress; (iii) both TI and FD were negatively correlated with the drought tolerance of the seedlings. Our results indicated that plants could adapt to drought by different strategies such as adjusting biomass allocation and root morphology, reducing root branch strength, and branching angles. Roots with narrower branching angles, greater branching complexity, larger TI, and consuming higher cost are more drought-tolerant.

**Keywords:** adaptive strategy; drought tolerance; root architecture

## 1. Introduction

Drought is one of the most prevalent agroforestry disasters and also a major factor restricting the development of forestry and agriculture in the world [1,2]. It has been estimated that global warming will cause more frequent and destructive future droughts [3–5]. Plants' tolerance to extreme drought varies greatly across species, determining vegetation types in regions with different drought degrees and whether plants survive extreme drought [6].

Root tissues play a major role in plant growth by exploiting soil resources via the absorption of nutrients and water [7,8]. Roots are able to respond to drought depending on a range of adaptive strategies including root distribution [9], root biomass adjustments [10], root physiological, and morphology plasticity [11] to give plants the ability to tolerate

or avoid drought stress. Root diameter, root branching angle (RBA), specific root length (SRL), root length density, and specific root area (SRA) are considered as the key root traits affecting water absorption capacity, plant productivity, or odds of survival under drought conditions [12]. Root phenotypic plasticity including increased ratio of root biomass distribution in deep soil [13], reduced vessel diameter of xylem to conserve soil water [14], reduced root angle for extracting water from deeper soil [15], and increased root–shoot ratio to improve water absorption across the soil layer can be used as indicators of the drought tolerance of plants [16]. Morphological plasticity and water-seeking ability of different plants under drought conditions are significantly species-specific, which may aggravate the inter-specific resource competition and species invasion, and the elimination of species with low tolerance causing reduced species diversity.

Researchers often define plant drought tolerance by analyzing the stability of plant biomass, physiology, and morphology under drought conditions. Biomass is used as a key indicator to judge drought tolerance because it concerns plants' growth capacity, forest carbon sink levels, and economic benefits of woody tree species [17]. Currently recognized intrinsic factors in determining plant drought tolerance include developmental stage, external morphology, drought escape, and water potential regulation ability [6,18], and so on. The basal morphology and morphological plasticity of roots are important parts of the intrinsic factors in plants. In relatively arid areas, drought-tolerant species tend to have adaptive advantages in their root architecture, such as tending to construct herringbone branching root or low branch density root patterns [19,20]. It has been suggested that "Steeper (root with a smaller branch angle), cheaper (root with a smaller diameter or a larger SRL) and deeper" roots are often the ideotype for plant survival in a persistently arid soil environment [21]. Plants with a well-developed taproot have a greater advantage in responding to drought compared to species with developed lateral roots because their taproots can reach deeper soil layers [22,23]. Root biomass and root morphology of species with a developed taproot are relatively stable under drought conditions [24,25]. Currently, plants' root architecture can be quantified by analyzing RBA and topological index (TI). Two representative root topologies have been reported [26], dichotomous branching and herringbone branching (Figure 1). Dichotomous branching roots can grow quickly in soils with scarce resources because their branching consumes less carbon. By contrast, herringbone branching roots grow slowly and consume more carbon, and they show higher drought tolerance than dichotomous branching roots in soils with a progressive water distribution because their taproots are developed [23,25–29]. However, the differences in drought-adaptive strategies exhibited by species with differences in drought tolerance and the key root architecture traits affecting drought tolerance are not clear.

Chinese fir (*Cunninghamia lanceolata* (Lamb.) Hook.), masson pine (*Pinus massoniana* (Lamb.)), and moso bamboo (*Phyllostachys edulis* (Carr.) H. de Lehaie f. edulis) are the three most widespread species in subtropical China that have contributed great value in the economy and culture [30,31]. They usually co-exist as a mixed forest and compete for resources, including water, light, and nutrients, etc. The frequency and intensity of droughts are predicted to become higher in subtropical areas [32]; therefore, species with strong water competitiveness or drought tolerance may dominate or even replace drought-intolerant species. Clarifying the drought tolerance and water competition strategies of co-existing species can help us to explore the competition mechanisms among species, which can be used as a theoretical basis for maintaining the stability and biodiversity of mixed forests. The three species exhibit highly different root morphologies, root architectures (moso bamboo and Chinese fir are fibrous root species, while masson pine are taproot species), and growth rates [33]. However, it is not known whether there are significant differences in the root adaptive strategies and drought tolerance among the three species. The seedling is a key stage in exploring root formation strategies of plants, as high morphological plasticity and high mortality under drought conditions often occur during the seedling stage [34]. The root is the primary developing organ of the three species in the seedling phase, while the shoot grows slowly [35,36]. Therefore, we measured seedlings of three species to

explore the drought-adaptive strategies of roots and evaluate the key root architecture traits affecting the drought tolerance of the three species. The hypotheses tested in this study are: (1) masson pine's drought tolerance will be significantly higher than that of moso bamboo and Chinese fir, and the root adaptive strategies of the three species will be species-specific; (2) TI, fractal dimension (FD), and root branching angle (RBA) of the three species change significantly under drought stress and these key traits are associated with drought tolerance significantly. The hypotheses will be verified by exploring the changes in the growth and root architecture parameters of the three species under drought, quantifying the drought tolerance, and analyzing the relationship between key architecture parameters and drought tolerance. Our study will address the knowledge gap of the relationship between plants' drought tolerance and root architecture.

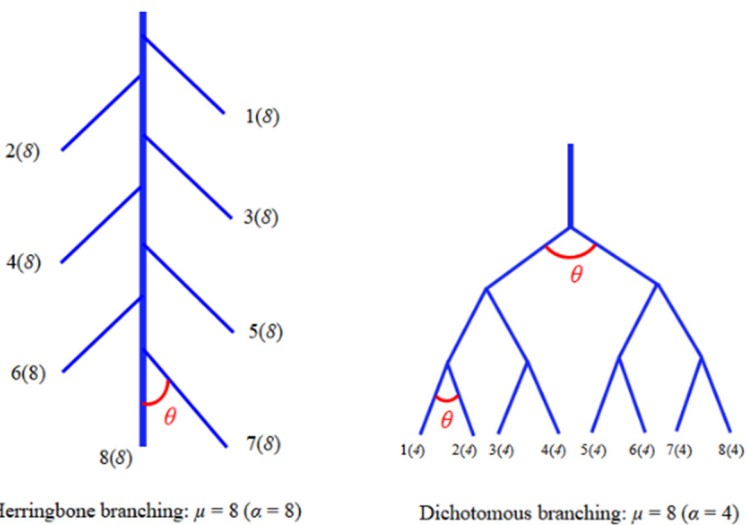

**Figure 1.** Diagram of root topology classification. $\alpha$ (altitude) is the number of internal links in the longest path from root collar to external tip, $\mu$ (magnitude) is the number of external links of the whole root system, and $\theta$ is the root branching angle. The TI of herringbone branching root and dichotomous branching root are 1 and 0.5, respectively. Herringbone branching root focuses on taproot development and consumes more costs, while dichotomous branching root focuses on lateral root development and consumes less costs.

## 2. Materials and Methods

### 2.1. Experimental Set Up

Three species were selected for this study: Chinese fir (*Cunninghamia lanceolata* (Lamb.) Hook.), masson pine (*Pinus massoniana* (Lamb.)), and moso bamboo (*Phyllostachys edulis* (Carr.) H. de Lehaie f. edulis)]. These species possess contrasting root growth strategies, for example, moso bamboo and Chinese fir roots belong to the dichotomous branching group (Figure 1) and focus more on the growth of their lateral roots in the topsoil, while masson pine belongs to the herringbone branching group and focuses more on the growth of its taproot.

This experiment was conducted in the Institute of Subtropical Forestry (119°95′ E and 29°48′ N), Hangzhou, Zhejiang Province. This location features a typical subtropical monsoon climate. The average annual sunshine duration, frost-free period, and the average temperatures in June, July, August, and annually were 1663.2 h, 307 d, 24.1 °C, 33.1 °C, 33.4 °C, and 17.8 °C, respectively, and the average relative humidity of the three months was 70.3%.

The parent plants of Chinese fir, moso bamboo, and masson pine were located at the Changle forestry center (Hangzhou, Zhejiang Province), Da Jing town forestry center (Guilin, Guangxi Province), and Laoshan forestry center (Hangzhou, Zhejiang Province), respectively. In March, three species' seeds soaked in deionized water were grown in a temperature incubator until germination. Seedlings with consistent bud length were

colonized in seedling disks. The substrate for filling the seedling disks was composed of peat, vermiculite, and perlite in the ratio of 2:1:1. In April, seedlings with the same height and basal diameter of each species were planted in plastic pots measuring 25 cm in diameter inside and 27 cm in height. Preliminary tests have verified that the container is large enough to support the unrestricted growth of all seedlings for three months. Every pot was filled with 6 kg of soil. The soil (red soil) that had never undergone fertilization, water control, and cultivation experiments was derived from forests in the Fuyang District of Zhejiang Province to simulate natural conditions for the three species' growth. One kilogram of the soil (pH = 4.91) contained 0.86 g of N, 11.2 g of K, 0.26 g of P, 85.13 mg of hydrolyzable N, 65.73 mg of available K, 4.15 mg of available P, and 18.7 g of organic carbon. The soil moisture content was controlled at 80%~85% of the maximum field water-holding capacity.

From June to August, the whole experiment was conducted in a greenhouse. Three levels of water supply were set in this study: the control (CK): 80%~85% of the maximum field water-holding capacity, moderate drought (M): 50%~55%, and severe drought (S): 30%~35%. The drought treatments were dried naturally to achieve the target soil moisture content. The soil weighing method was employed to control soil moisture every day and a soil moisture measurement system (IMKO micromodtechnik GmbH Instrument, Trime-pico AZS-100, Ettlingen, Germany) was used to monitor the soil moisture every day. First, 6 kg of soil was dried at 105 °C to obtain its dry weight (DWs). We then calculated the total weight of the pots for each treatment when the preset soil moisture was achieved.

$$TWp = DWs \, (WCt + 1) + WP \tag{1}$$

where TWp is total weight of pot culture, WCt is preset soil moisture content of treatments (CK, M, or S), and WP is the weight of the empty pot. The TWp for each treatment is a fixed interval calculated from Equation (1). We brought the total weight of the pots to TWp by adding water to the soil moisture content of each treatment within its predetermined range. Pots of each treatment were weighed daily at 6 pm and then replenished with the water.

### 2.2. Harvest and Measurements

Three samplings were performed over a 90-day test period. The interval of each sampling was 30 days. The first sampling (June) occurred 30 days after the soil moisture content of each treatment reached the preset level. Five plants of each species were harvested for each sampling. A total of 45 seedlings were harvested by the end of the experiment.

The shoots and roots were separated at the soil surface with sharp scissors. The soil on the root surface was carefully shaken off to avoid breaking the root tissue and root architecture. The remaining roots in the soil were collected by screening the soil with a 2 mm sieve.

The washed roots were scanned by a double-sided scanner to obtain root images with a resolution of 500 dpi. Root architecture (FD and RBA) and root morphology (root length (RL), root surface area (RSA), number of root links, and root average diameter (RD)) were obtained by analyzing the root images with the WinRhizo software (Regent Instruments Inc., Quebec, WinRhizo Pro, Canada). The analysis of FD was performed by the WinRhizo software referring to box-counting method [26,37]. The software simulation covered the image using a large number of empty square boxes with a consistent side length ($\varepsilon$) and recorded how many (N) boxes were not empty and then calculated FD using N and $\varepsilon$. The leaves, stems, and roots were devitalized at 105 °C for 30 min to cease their physiological metabolism and dried at 65 °C to a constant weight to obtain the dry weight of the three parts (biomass) [38].

### 2.3. Calculations and Statistics

The response ratio (RR) was the ratio of CK′ and Tj, which was used to estimate quantitatively the response of seedling growth parameters to drought treatments. CK′

represents the average value of the control group and Tj represents the values of five replicates (j = 1, 2, ... , 5) of each treatment.

$$RR = (CK' - Tj)/CK' \tag{2}$$

$$TI = \log(\alpha)/\log(\mu) \tag{3}$$

The root tips and branch points were termed as external and internal nodes, respectively, and a root segment between the two nodes was termed as a link. Links that did not terminate at both ends were called internal links, while those that terminate at one end were called external links. As shown in Equation (3), TI (root topological index) is the ratio of lg ($\alpha$) and lg ($\mu$), where $\alpha$ (altitude) is the number of internal links in the longest path from root collar to external tip and $\mu$ (magnitude) is the number of external links of the whole root system [23].

Statistical analyses and correlations between parameters were analyzed using the SPSS software. Significant differences in the data between different drought intensities and different drought durations were tested using factorial analysis of variance (ANOVAs). The correlations between root architecture parameters (TI, RBA, FD) and RR were analyzed using Pearson's correlation analysis. The residual normality of raw data for each parameter (TI, RBA, FD) was assessed via the Shapiro–Wilk test before statistical tests.

## 3. Results

### 3.1. Effect of Drought on Biomass Accumulation and Allocation

The dry weight of the root, shoot, and the root–shoot ratio were measured to reflect the growth characteristics and carbon allocation strategies of the three species under drought treatments. Compared with CK, M and S tended to decrease the biomass of the root and shoot (Figure 2). M increased the root–shoot ratio of the three species significantly. S only increased the root–shoot ratio of Chinese fir significantly, while it did not affect the root–shoot ratio of bamboo and pine (Figure 2).

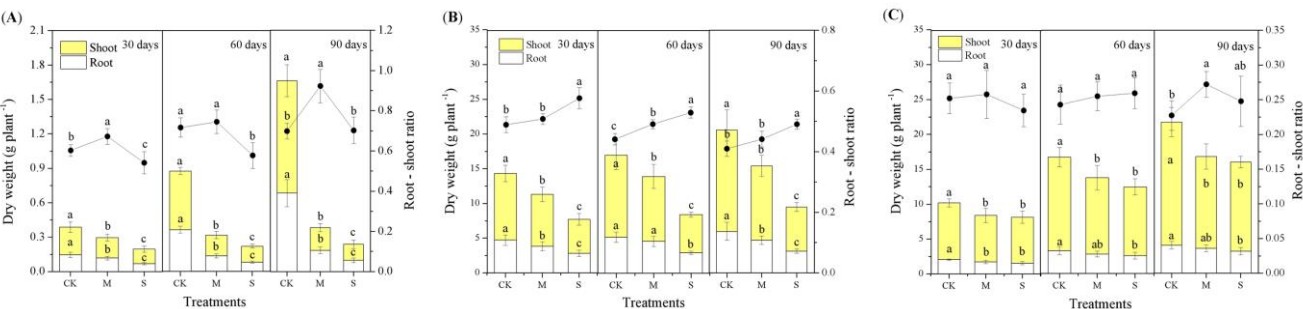

**Figure 2.** Effects of different drought treatments on dry weights (column charts) and root–shoot ratios (line charts) of the three species during three sampling periods. The different lowercase letters denote significant differences among treatments (*p* < 0.05). (**A**): moso bamboo, (**B**): Chinese fir, (**C**): masson pine. CK: the control, M: moderate drought, S: severe drought.

### 3.2. Effect of Drought on Root Morphology

RL, RSA, and RD were measured to reflect the ability of elongation, absorption area extension, and radial growth of the root under drought treatments. The RL and RSA showed a decreasing order as Chinese fir > pine > bamboo (Figure 3, Table A1). Compared to CK, M and S tended to decrease the RL and RSA (Figure 3) for the three species. For moso bamboo, severe drought was more significant in reducing the root length and surface area than moderate drought, but not for masson pine and Chinese fir.

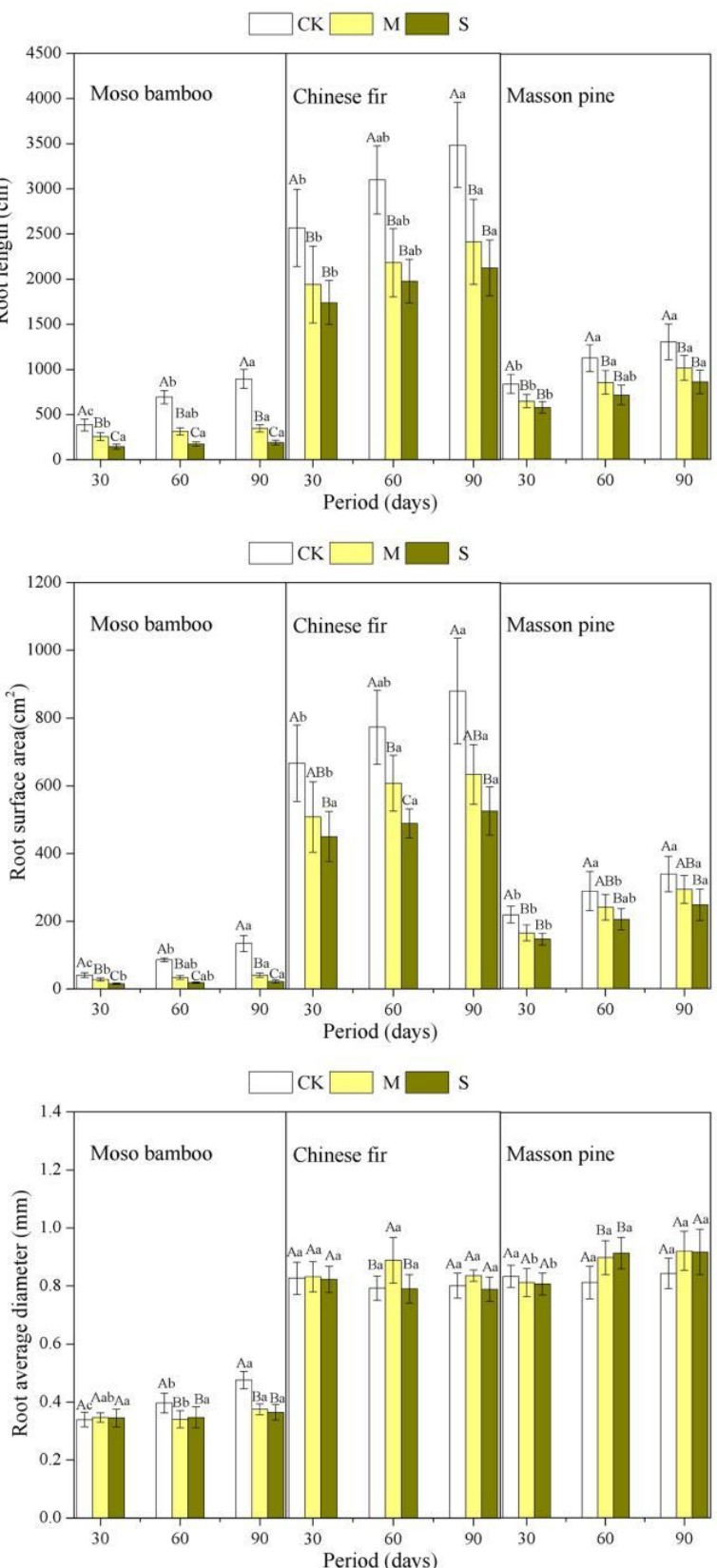

**Figure 3.** Effects of different drought treatments on root growth of the three species during three sampling periods. The capital letters denote significant differences among treatments (*p* < 0.05), and the lowercase letters denote significant differences among periods (*p* < 0.05). CK: the control, M: moderate drought, S: severe drought.

The response ratios of biomass (RRB) and root length (RRL) to drought were used to evaluate the drought tolerance of the plants. Both RRL and RRB showed a decreasing order as bamboo > Chinese fir > pine ($p < 0.05$) (Figure 4, Table A1). The response ratios to S for root length and root biomass of bamboo and root length of Chinese fir were significantly greater than that to M (Figure 4).

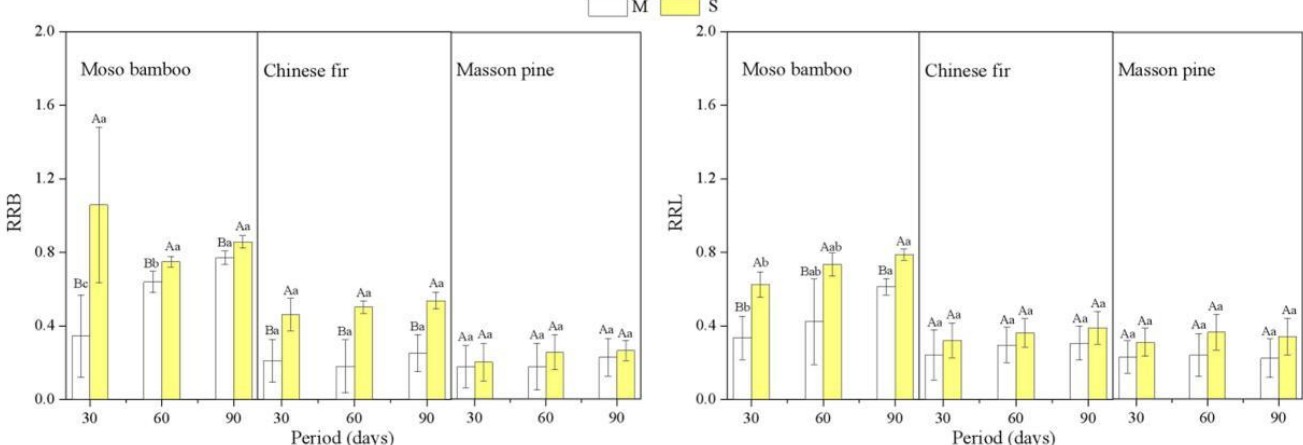

**Figure 4.** Response ratios of biomass (RRB) and root length (RRL) for the three species to different drought treatments. The capital letters denote significant differences among treatments ($p < 0.05$), and the lowercase letters denote significant differences among periods ($p < 0.05$).

The RD values of the three species were not significantly affected by drought during the first 30 days. Drought reduced the RD of moso bamboo, while increasing the RD of masson pine at the 60th day. M increased Chinese fir's RD significantly, but severe drought did not (Figure 3).

### 3.3. Effect of Drought on Root Architecture

TI showed a decreasing order as masson pine > Chinese fir > bamboo (Figure 5, Table A1). Drought increased the TI of all the three species (Figure 5). The TI of bamboo and pine increased gradually with the increase in drought. For Chinese fir, the TI under M was higher than that under S. Furthermore, S did not increase masson pine's TI significantly until 60 days later. Drought caused a decrease in the RBA and FD of Chinese fir and bamboo, but not for masson pine.

### 3.4. Association of RR with Root Architecture Parameters

Both TI and FD were negatively correlated with RRL and RRB ($p < 0.01$). No significant correlation was observed between the RBA and RR (RRL and RRB) (Figure 6).

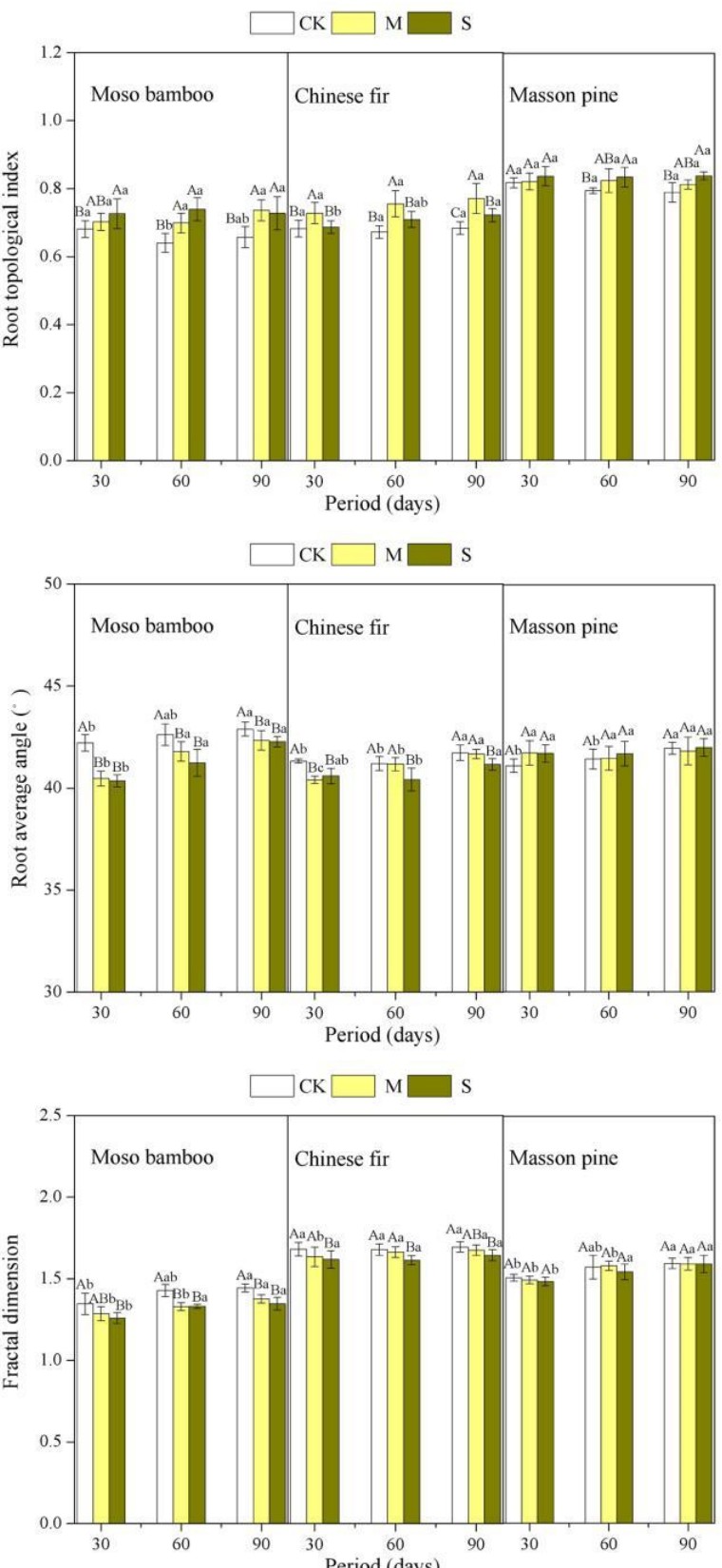

**Figure 5.** Effects of different drought treatments on the root architecture of the three species during three sampling periods. The capital letters denote significant differences among treatments ($p < 0.05$), and the lowercase letters denote significant differences among periods ($p < 0.05$).

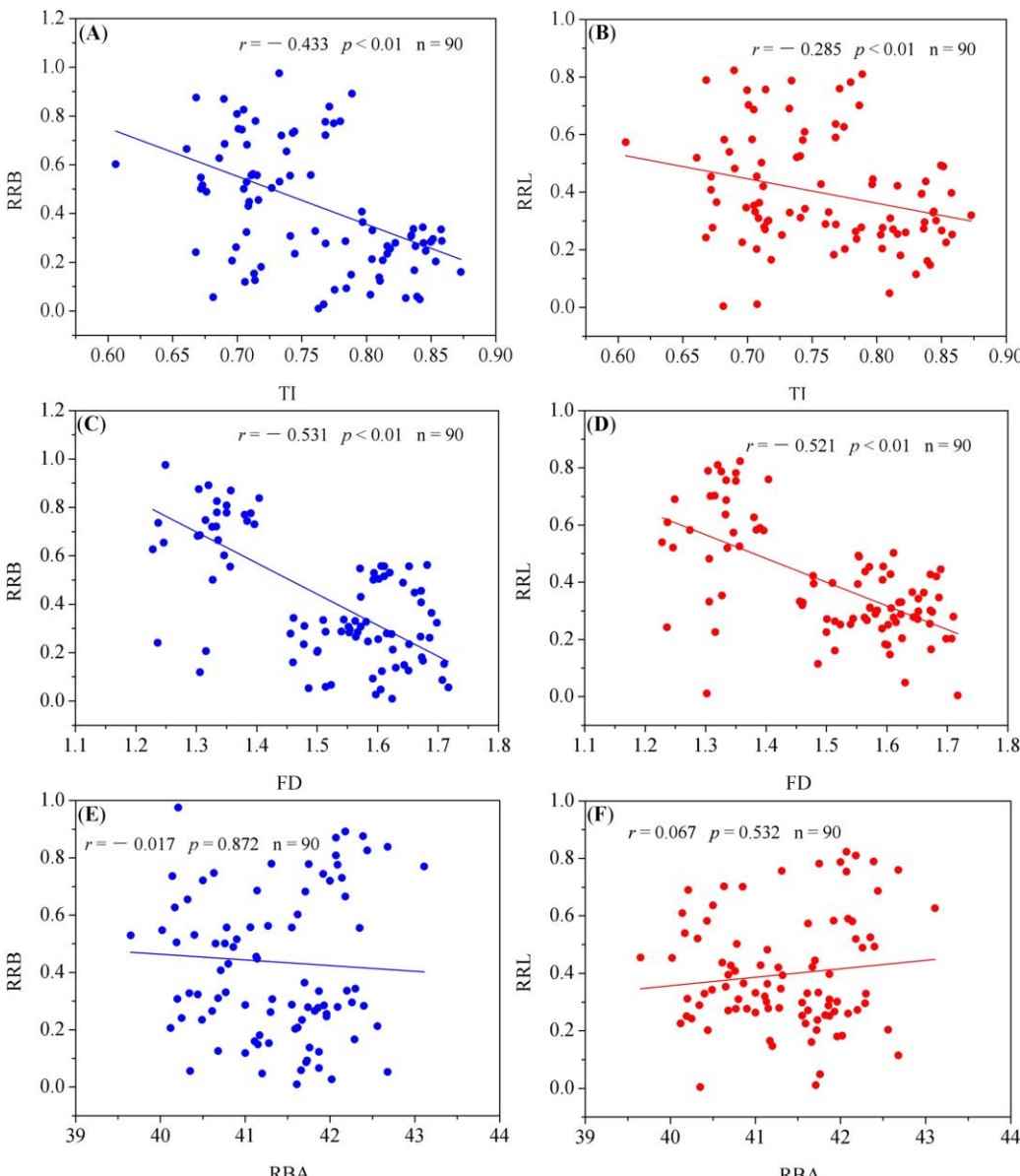

**Figure 6.** Relationships between RR and root architecture parameters (TI, RBA, FD). r: correlation coefficient, n: total number of samples. The relationship between RRB and root architecture parameters ((**A**): TI, (**C**): FD, (**E**): RBA) is represented by three blue scatter diagrams, while the relationship between RRL and root architecture parameters ((**B**): TI, (**D**): FD, (**F**): RBA) is represented by three red scatter diagrams.

## 4. Discussion

### 4.1. Response of Root Growth to Drought

In this study, drought tolerance showed a decreasing order as pine > Chinese fir > bamboo. Different levels of drought inhibited the biomass accumulation and root growth of the three species, similar to previous studies [39,40]. Plants close their stomata to prevent water loss under drought conditions, reducing carbon dioxide assimilation rates and photosynthesis products [41]. The reduction in biomass is not just a drought-induced decline in carbon accumulation, but it can also be considered as an adaptive strategy for plants. For example, plants reduce the water transpiration of their leaves by reducing aboveground growth and promoting the extension of primary roots to bring the roots to deeper and moist soil layers by reducing the growth of lateral roots in response to drought [40,42]. The three species increased their root–shoot ratios under treatment M,

indicating that they exhibited the same carbon allocation strategy of increasing carbon input to their roots in response to drought, similar to the results in other species [43]. Under drought stress, shoot growth was severely inhibited, while root growth continued through essential reserve translocation from the shoot using long-distance chemical and hydraulic signal transduction [18,44]. Studies have previously found a significant correlation between the growth of roots and shoots under drought conditions, which was thought to be a coordination strategy of plant growth [45,46]. However, the trends of the root–shoot ratios of moso bamboo and pine were not consistent over time (Figure 2). A possible explanation is that a clash occurred between the growth of the roots and shoots at a critical developmental stage [47] (the second 30 days in this experiment) and appeared to be exacerbated by drought-induced carbon limitation.

Overall, the growth of all three species was inhibited by drought, but the degree of inhibition varied significantly (Figure 3). This indicates that drought tolerance varied significantly among the three species, with bamboo being the most vulnerable and pine the least. Moreover, the morphological plasticity of the three species was also different in drought conditions, such that bamboo tended to proliferate finer roots, while the pine tended to proliferate thicker roots. These results revealed that the radial growth of the three species' roots showed different adaptive strategies and was significantly species-specific under different drought conditions. Proliferation of thinner roots of perennial grasses under drought conditions has been found in previous studies [48], because thinner roots require less carbon for morphological construction and respiration consumption and reach further soil space [49,50]. Plant root thickening under drought conditions also has been reported [51–54]. Plants reduced the number of lateral and axial roots [51,52] and increased root crown width [53] and root thickness [54] in response to the increase in penetration resistance of the uppermost soil layer caused by drought conditions.

### 4.2. Response of Root Architecture to Drought

From our study, all the three species were found to increase their TI to respond to drought (Figure 5). This is the result of plants suppressing lateral root development and preferring to develop their taproots under drought conditions. A similar result was found that perennial grasses increased their TI after 15 days of drought stress [48]. Lynch also found that the strategies of increasing TI helped plants absorb water in arid soil [42]. Another explanation for increasing the TI is that this optimizes the carbon allocation. It has been suggested that lateral root respiration will consume more carbon [55,56], so reducing lateral roots is more cost-effective for plants in drought. In addition, Gao and Lynch found that plants attenuated internal competition in the root system and devoted more resources (photosynthates and water) to form taproots and deeper roots by limiting the branches of lateral roots [57]. The FD of the bamboo and Chinese fir's roots reduced significantly under drought stress in agreement with the above viewpoint. The decrease of FD indicated that root complexity decreased and root branch development was inhibited by drought. This is an adverse reaction under drought and is also an adaptive strategy. The relatively simple root architecture has a survival advantage in arid soil over complex roots [58]. Zhan found that reducing the branch density of lateral roots improved the drought tolerance of maize [59].

"Steeper and deeper" architectures are considered as the ideal root traits with high water absorption efficiency [21]. Manschadi also found that drought-tolerant wheat genotypes generally had narrower root branches than drought-sensitive wheat genotypes [15]. However, no researchers have explored whether the root branching angle of plants is changed by drought stress, or discussed the change as an adaptive strategy of plants under drought. In this study, RBA was used to reflect directions (horizontal or longitudinal) of root growth and water-seeking. We found that moso bamboo and Chinese fir tended to reduce the RBA under drought stress, while pine showed no obvious response (Figure 5). It has been found that abscisic-acid-mediated root hydrotropism caused changes in the direction of root growth for *Arabidopsis* under drought stress [60]. Root plasticity is not

only reflected in quantity and branching mode, but also in the water-seeking direction of root systems. Root branching angles in drought-tolerant species may be stable under drought conditions, for example pine. The first hypothesis that masson pine's drought tolerance will be significantly higher than that of moso bamboo and Chinese fir and the root adaptive strategies of the three species will be species-specific was verified by the three tree species exhibiting differences in drought tolerance and plasticity of root morphology and architecture under drought conditions.

### 4.3. Relationship between Root Architecture and Drought Tolerance of Species

From this study, it was found that the growth, root morphology, and root architecture of masson pines were more stable under drought conditions compared with the other two species. We quantified the extent that root growth and biomass accumulation responded to drought with RR and found that drought tolerance of the three species showed a decreasing order as pine > Chinese fir > bamboo. We also evaluated which root characteristics were the key factors in determining a plants' drought tolerance by analyzing the correlation between the indicators of root architecture and the RR (Figure 6). The results proved that individuals with larger root TI and FD were more drought-tolerant. From the results of the correlation analysis, no significant correlation was found between the RBA and RR. In fact, Chinese fir and pine with smaller RBA values were more drought-tolerant relative to bamboo. RBA is considered by many researchers as an important adaptive trait for plants under prolonged drought conditions [18]. Although there is no direct evidence that plants with a small RBA are more drought-tolerant, some studies have shown that the RBA is significantly associated with the depth that the roots can reach [61,62]. Species with a small RBA may conserve energy to support the roots to penetrate to the deeper soil horizons [63], and can efficiently capture water from the soil which is dry in the surface layer and wet in the deep layer [6,64]. The second hypothesis that the TI and FD of the three species changes significantly under drought stress and these key traits are associated with drought tolerance significantly was also justified by the changes in root architecture traits of the three species under drought conditions and the relationship between root architecture and drought tolerance.

The results of TI indicated that species with developed taproots and fewer lateral roots were more drought-tolerant. The length of the taproot usually represents the soil depth where the plant roots can reach. Similar results were found that cultivars possessing longer taproots and lower lateral root branching density were more likely to access water from deep soil layers under drought conditions [46]. Lynch reported that reducing root branching facilitates crop survival in drought conditions [42]. Reduced axial roots, lateral root density, and the loss of carbon caused by growing useless roots were considered to be adaptive strategies of plants to improve water capture [42].

The more complex (the larger FD) the plant root branches are, the stronger their drought tolerance is, which seems to be contradictory with the knowledge that plants with a more simplified architecture are more drought-tolerant. It can be understood that plants with a developed primary root and a large number of branches on it, with more lateral roots but less secondary branches are the more ideal roots [65,66]. Root branches are very necessary for plants under drought, but preferably occur on the primary roots that can reach the deeper soil layer rather than on the lateral roots in shallow soil. Plants have to allocate their limited carbon to taproots that can better absorb water from the deep soil layer under drought conditions, resulting in root system simplification. As a result, plants distributing expensive lateral roots in the surface layer are not conducive to their drought tolerance [64]. Combining the results of TI and RBA, a drought-tolerant root should be narrower, with a developed taproot, and a large number of secondary branches on the taproot, not just elongation. Roots use limited carbon as much as possible for their taproot elongation and branching to cope with drought, which is also a manifestation of the optimized carbon allocation [13].

## 5. Conclusions

Drought with different intensities inhibited the biomass accumulation and root growth of the three species. However, the three species showed different morphological and architectural plasticity and drought tolerance under drought conditions. Bamboo tended to proliferate finer roots in response to drought, and pine was relatively stable. Both bamboo and Chinese fir tended to reduce their root branch strength and root branching angle under drought stress, while pine was relatively stable. The common denominator was that the roots of all the three species tended to develop a herringbone branching architecture (increase their TI) under drought stress. We quantified the extent that root growth and biomass accumulation responded to drought with the response ratio and found that the drought tolerance of the three species showed a decreasing order as pine > Chinese fir > bamboo. The importance of different root architecture traits for their drought tolerance was assessed. It is suggested that roots with narrower branching angles, greater branching complexity, larger TI, and higher cost are more drought-tolerant.

**Author Contributions:** Z.Y. and B.Z. designed the experiments; Z.Y., J.Z., H.N. and H.W. performed the analysis; Z.Y., B.Z. and J.Z. drafted the manuscript; All the authors critically revised and approved the final version of this manuscript. All authors have read and agreed to the published version of the manuscript.

**Funding:** This study was supported by funding from the National Natural Science Foundation of China (Nos. 32201645, 31670607 and 32071756), National Key R&D Program of China (2021YFD2200402), Forestry Science and Technology Project of Zhejiang Province (2022SY16).

**Data Availability Statement:** The data presented in this study are available on request from the corresponding author.

**Conflicts of Interest:** The authors declare no conflict of interest.

## Appendix A

**Table A1.** Effects of species, periods, treatments, and their interactions on growth and response ratio.

| Parameter | Treatment | | Period | | Species | | Period × Treatment | | Species × Treatment | | Species × Period | | Treatment × Species × Layer | |
|---|---|---|---|---|---|---|---|---|---|---|---|---|---|---|
| | *F*-Value | *p*-Value | *F*-Value | *p*-Value | *F*-Value | *p*-Value | *F*-Value | *p*-Value | *F*-Value | *p*-Value | *F*-Value | *p*-Value | *F*-Value | *p*-Value |
| Dry weight | 86.933 | <0.01 | 91.296 | <0.01 | 954.73 | <0.01 | 3.837 | <0.01 | 24.457 | <0.01 | 28.433 | <0.01 | 0.476 | 0.871 |
| TI | 33.828 | 0.031 | 1.125 | 0.328 | 215.192 | <0.01 | 2.402 | 0.054 | 7.284 | <0.01 | 2.112 | 0.084 | 0.812 | 0.594 |
| FD | 24.379 | <0.01 | 39.885 | <0.01 | 715.003 | <0.01 | 0.177 | 0.950 | 4.757 | <0.01 | 4.619 | <0.01 | 0.4 | 0.863 |
| RBA | 19.831 | <0.01 | 47.610 | <0.01 | 34.957 | <0.01 | 1.585 | 0.183 | 14.406 | <0.01 | 7.072 | <0.01 | 2.853 | <0.01 |
| RRB | 51.506 | <0.01 | 2.633 | 0.079 | 109.523 | <0.01 | 4.014 | <0.05 | 7.870 | <0.01 | 0.291 | 0.883 | 6.092 | <0.01 |
| RRL | 44.689 | <0.01 | 6.930 | <0.01 | 74.610 | <0.01 | 0.296 | 0.745 | 6.520 | <0.01 | 2.784 | 0.033 | 0.547 | 0.702 |

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
