# Peer review of "Three Subtropical Species Adapt to Drought by Reallocating Biomass and Adjusting Root Architecture"

_forests, doi:10.3390/f14040806_

Round 1

Reviewer 1 Report

Forests-2252002 – Review

 General comment:  This is a study focused on root systems’ response to drought.  Roots are understudied in general, making the results of this research of use to many interested in how plants respond to drought.

Title:  One of the three species is not a tree but a grass.  May need to change ‘tree’ here with ‘species’ as well as throughout the document.

Abstract:  Phyllostachys edulis is a grass (bamboo) – it is not a tree.

 Key words:  Several words are in the title and could be removed; not alphabetized.

 Introduction:  Many tense and wording choice errors, likely due to English not being the authors’ first language.  Any revision will require a great deal of editing.  In some cases the language issues make the writing difficult to interpret.  For instance, what is meant by ‘otherness’ on line 82? 

 It sounds like the three species were chosen because they grow together but are known to have different root architectures or water acquisition strategies.  If that is the case, I’m not sure the first hypothesis is a scientific test but rather a confirmation of what is known about existing differences of the species.  Are the authors saying the different morphologies etc of the species will remain different under drought conditions?  If so, this needs to be clarified.  It is concerning that the authors called a grass (albeit a large grass – but still a monocot vs two gymnosperms) species a tree – makes me as a reviewer question their familiarity with the species. 

 Do climate change models predict drought in this subtropical region where these three species coexist?  It is not clear why these species, other than they are different and are found together, were chose to study drought tolerance.  More background is needed.

 Materials and Methods:  How were the 1-yr old seedlings produced/grown? Three forestry centers were listed – did one species come from each or all species from each? Under what conditions were these species grown for a year? For the greenhouse experiment using the 1-yr old seedlings, at what dept was the field soil collected?  What was the soil type?  What is meant by ‘clear’ water?  The presentation of Figure 1 seems odd.  To which of the two do each species belong?

 Design and analyses are robust; though more details about any necessary data transformations should be provide.

 Results: It is not clear why Figure 4 and the analyses associated with this figure did not include the control plants for comparison. Otherwise good presentation.

 Discussion and Conclusions:  It would be better if this section was organized around the two hypotheses – clearly stating if each were met or not.  The main finding seems to be that plants with tap roots will survive better under drought conditions – but didn’t we already know that?

Author Response

Response to Reviewer 1 Comments

Thank you very much for your comments. We have revised and clarified all the errors and confusions you pointed out.

Q1:Title: One of the three species is not a tree but a grass. May need to change ‘tree’ here with ‘species’ as well as throughout the document.

Response: Per your suggestion, we revised the title to “Three subtropical species adapt to drought by reallocating biomass and adjusting root architecture”.

Q2:Abstract: Phyllostachys edulis is a grass (bamboo) – it is not a tree.

Response: Per your suggestion, we have revised this error.

Q3: Several words are in the title and could be removed; not alphabetized.

Response: One keyword was removed and all keywords were alphabetized.

Q4: Introduction: Many tense and wording choice errors, likely due to English not being the authors’ first language. Any revision will require a great deal of editing. In some cases the language issues make the writing difficult to interpret. For instance, what is meant by ‘otherness’ on line 82?

Response: Per your suggestion, we have re-edited the full manuscript to make the statement easier to understand. We focused on modifying the incorrect tenses and words.

Q5: Introduction: It sounds like the three species were chosen because they grow together but are known to have different root architectures or water acquisition strategies. If that is the case, I’m not sure the first hypothesis is a scientific test but rather a confirmation of what is known about existing differences of the species. Are the authors saying the different morphologies etc of the species will remain different under drought conditions?  If so, this needs to be clarified. It is concerning that the authors called a grass (albeit a large grass – but still a monocot vs two gymnosperms) species a tree – makes me as a reviewer question their familiarity with the species.

Response: The ambiguity was caused by our unclear writing, which has been corrected. We have also clarified the three reasons for being selected. see line 85-98. In fact, it is not known whether there are significant differences in the root adaptive strategies and drought tolerance among the three species. Therefore, we propose the first hypothesis.

The three species were selected for the following reasons. The three species exhibit highly different root morphology, root architecture (moso bamboo and Chinese fir belong to the fibrous root species, while masson pine belongs to the taproot species) and growth rates. In addition, three species usually co-exist as mixed forest and are competing for resources, including water, light, and nutrients, etc. The species among them with strong water competitiveness or drought tolerance may dominate or even replace the weak species in the competition. Clarifying the drought tolerance and water competition strategies of coexisting species can help us to explore the competition mechanism between species, which can be used as a theoretical basis for maintaining the stability and biodiversity of mixed forest.

Q6: Do climate change models predict drought in this subtropical region where these three species coexist? It is not clear why these species, other than they are different and are found together, were chose to study drought tolerance. More background is needed.

Response: Based on the analysis of meteorological data from 1960 to 2000, a study reported that the frequency and severity of drought in subtropical areas will improve. Per your suggestion, we added the relevant background. See line 89-90.

Q7: Materials and Methods: How were the 1-yr old seedlings produced/grown? Three forestry centers were listed – did one species come from each or all species from each? Under what conditions were these species grown for a year? For the greenhouse experiment using the 1-yr old seedlings, at what dept was the field soil collected? What was the soil type? What is meant by ‘clear’ water? The presentation of Figure 1 seems odd. To which of the two do each species belong?

Response: Per your suggestion, we added the relevant background and modified the incorrect words. See line122-123,131-143. 

Moso bamboo and Chinese fir roots belong to the dichotomous branching (Figure 1) and focus more on the growth of their lateral roots in the topsoil, while masson pine belongs to the herringbone branching and focuses more on the growth of its taproot.

The soil (red soil) was derived from forests in Fuyang District of Zhejiang Province to simulate natural conditions for three species’ growth.

In addtion, one species come from each forestry center. The parent plants of Chinese fir, moso bamboo and masson pine are located at Changle forestry center (Hangzhou, Zhejiang Province), Da Jing town forestry center (Guilin, Guangxi Province) and Laoshan forestry center (Hangzhou, Zhejiang Province) respectively.

Q8: Results: It is not clear why Figure 4 and the analyses associated with this figure did not include the control plants for comparison. Otherwise good presentation.

Response: Because the values of RRL and RRB were calculated from Equation 2. The RR represented the gap between the treatment group and the control. RR of control should be 0, meaningless for the results. Therefore, we did not show the RR of the control group.

Q9: Discussion and Conclusions: It would be better if this section was organized around the two hypotheses – clearly stating if each were met or not. The main finding seems to be that plants with tap roots will survive better under drought conditions – but didn’t we already know that?

Response: Per your suggestion, we revised the conclusions and discussions following the hypotheses.

Reviewer 2 Report

The manuscript was reviewed. It presents a series of evaluation on three "contrasting" species typically found in China's subtropical forests to elucidate their response to drought.

The manuscript is well written, clear, with just some minor spells errors. The results are clearly presented and I think they are of wide interest for the readership. 

Overall, the manuscript is publishable just after some minor revisions. You can see my suggestions and comments in the attached pdf file REVIEWED.

Author Response

Response to Reviewer 2 Comments

Thank you very much for your comments. I have revised the errors in grammar, words and graphs according to your opinion, and added the necessary background and information. Please check the uploaded manuscript with traces of modification.

Round 2

Reviewer 1 Report

The authors have addressed my concerns.  There are still a few typos and language issues.  For example: line 14 -- best not to start a sentence with 'But'  -- would combine with previous sentence. Line 231 -- each figure caption needs to stand alone -- don't write 'The same below'  -- not a complete sentence and not informative -- what is the same and where below?  Line 132 -- should be 'Preliminary'.  Line 269 -- combine sentence starting with 'And' with previous sentence.

Author Response

Thank you for your comments. I have revised the manuscript according to your opinion.
